# Investigation of Oil Spills from Oil Tankers through Grey Theory: Events from 1974 to 2016

**Dong-Taur Su [1], Fu-Ming Tzu [2,*] and Chung-Hung Cheng [3]**

[1]  Department of Shipping Technology, National Kaohsiung University of Science and Technology, Kaohsiung 80543, Taiwan; tonysu@nkust.edu.tw

[2]  Department of Marine Engineering, National Kaohsiung University of Science and Technology, Kaohsiung 80543, Taiwan

[3]  Maritime Patrol Directorate General, Coast Guard Administration, Executive Yuan, Southern Sector Flotilla, Xinbei 25152, Taiwan; cchung0704@gmail.com

*  Correspondence: fuming88@nkust.edu.tw, Tel.: +886-7-810-0888 (ext. 25245)

**Abstract:** An oil spill from a vessel is a critical maritime accident that can severely damage the environment. In this study; we utilize the basic construction of grey relational analysis to explore oil spill events statistics from 1974 to 2015 and successfully analyze the causes of incidents in 2016. The results illustrate that grey relational analysis effectively identifies the factors causing oil spills with an accuracy of over 96%. The research is aimed to reduce the marine accidents and predict the cause of oil spill in advance. The analysis is dealing with the incidents to approach the circumstance in various intensity of oil spill in the last 40 years. Moreover, an application of grey theory demonstrates accurate and reliable methodology to decision maker. Thus, the investigation can predict the causes of pollution from oil spill accidents in the future.

**Keywords:** oil spill; oil tanker; investigation event; grey relational analysis

## 1. Introduction

Numerous countries have made great efforts towards ecological protection for many years. However, an unavoidable oil spill accident from an oil tanker is a critical event and harmful to the environment. Here, we investigate the causes of oil spills to analyze the related factors using grey theory. Among the worst accidents of oil spills from oil tankers was the Torrey Canyon incident that occurred in 1967 when the ship struck Pollard's Rock on the Seven Stones reef in the English Channel [1–3]. Consequently, the ship leaked approximately 119,000 tons of crude oil. Moreover, the Amoco Cadiz incident in 1978 led to a major oil spill near the coast of France, with a serious leakage of 223,000 tons of crude oil from the oil tanker [4–6]. Such incidents caused irreparable pollution of the ecological environment and are among the most unforgettable of marine events. In particular, the largest oil spill was from the Atlantic Empress oil tanker in an accident near the island of Tobago in the West Indies in 1979 [7]; the Atlantic Empress leaked more than 287,000 tons of crude oil that caused major damage to the marine environment. Zafirakou et al. [8] utilized "multi-criteria analysis of different approaches to protect the marine and coastal environment from oil spills" to discuss the marine pollution caused by oil spill of oil tanker. Taking the responsibility from local authorities and the relevant unit in case of an oil spill accident, through a multi-criteria methodology.

The United Nations officially adopted the United Nations Convention on the Law of the Sea (UNCLOS) in 1982 [9–11]. Presently, 123 countries have accepted this constitution of the oceans. After UNCLOS included the concept of exclusive economic zones (EEZs), the signatories delimited their EEZs and limited the economic activities of other countries in their EEZs [12–14]. Furthermore,

the European Union (EU) banned single-hull oil tankers from ocean shipping in 2000, modifying the International Convention for the Prevention of Pollution from Ships (MARPOL) [15,16]. As a result, the regulation led to the effective abandonment of single-hull oil tankers. Since 2010, MARPOL has stipulated that single-hull oil tankers should no longer be used in shipping. Although some shipping companies have continued to use single-hull oil tankers, some countries have banned single-hull oil tankers from entering their territories, such as the United States, Europe, Singapore, South Korea, and China. Meanwhile, extreme weather has frequently increased the intensity of tropical storms such as typhoons, hurricanes, and cyclones [17–19]. Therefore, maritime incidents remain a high risk for all ships at sea, without exception. Overall, 50% of shipping accidents have been caused by adverse weather [20–23].

The paper utilizes grey theory to examine the oil spills from oil tankers based on data from the International Tanker Owners Pollution Federation Ltd. (ITOPF) and investigates the incidence of oil spills using the information, as shown in Table 1 [24].

**Table 1.** Major oil spills since 1967 (quantities rounded to the nearest thousand).

| Year | Ship Name | Location | Spill Size (Tons) |
|------|-----------|----------|-------------------|
| 1967 | TORREY CANYON | Scilly Isles, UK | 119,000 |
| 1972 | SEA STAR | Gulf of Oman | 115,000 |
| 1975 | JAKOB MAERSK | Oporto, Portugal | 88,000 |
| 1976 | URQUIOLA | La Coruna, Spain | 100,000 |
| 1977 | HAWAIIAN PATRIOT | 300 nautical miles off Honolulu | 95,000 |
| 1978 | AMOCO CADIZ | Off Brittany, France | 223,000 |
| 1979 | ATLANTIC EMPRESS | Off Tobago, West Indies | 287,000 |
| 1979 | INDEPENDENTA | Bosphorus, Turkey | 94,000 |
| 1980 | IRENES SERENADE | Navarino Bay, Greece | 100,000 |
| 1983 | CASTILLO DE BELLVER | Off Saldanha Bay, South Africa | 252,000 |
| 1985 | NOVA | Off Kharg Island, Gulf of Iran | 70,000 |
| 1988 | ODYSSEY | 700 nautical miles off Nova Scotia, Canada | 132,000 |
| 1989 | KHARK 5 | 120 nautical miles off Atlantic coast of Morocco | 70,000 |
| 1989 | EXXON VALDEZ | Prince William Sound, Alaska, USA | 37,000 |
| 1991 | ABT SUMMER | 700 nautical miles off Angola | 260,000 |
| 1991 | HAVEN | Genoa, Italy | 144,000 |
| 1992 | AEGEAN SEA | La Coruna, Spain | 74,000 |
| 1992 | KATINA P | Off Maputo, Mozambique | 67,000 |
| 1993 | BRAER | Shetland Islands, UK | 85,000 |
| 1996 | SEA EMPRESS | Milford Haven, UK | 72,000 |
| 2002 | PRESTIGE | Off Galicia, Spain | 63,000 |
| 2007 | HEBEI SPIRIT | South Korea | 11,000 |
| 2018 | SANCHI | Off Shanghai, China | 113,000 |

## 2. Methodology

Grey theory is one of numerous applications for forecasting problems. The method is a model construction of grey relational analysis when the system is unclear and the information is incomplete. Grey relational analysis is effective in dealing with the cause of a problem in circumstances of uncertainty, multiple inputs, discrete data, and incomplete data [25,26]. In this study, the data on oil spills between 1974 and 2015 were analyzed. Based on the original data, a grey differential equation model was used. The investigation of grey relational analysis is based on the GM (1, 1) model. The method for existing data is actually to find out the dynamic behavior of each element in series.

GM (1, 1) indicates a first-order differential equation. The input variable is the one used for prediction of the problem. GM (1, N) indicates a first-order differential equation, where the number of input variables is N, as generally used in analysis of a multivariate problem. GM (0, N) is a special case of GM (1, N), which means a zero-order differential equation with N input variables, which is

used for multivariate analysis. In short, our task utilizes the form of GM (1, 1) to investigate the grey grade. The original data are standardized, and the maximum value in the sequence is used as a reference value. All values are between 0 and 1 and can be compared with one another. However, dispersed information cannot be used for comparison. Thus, the averaged value of the grey relational factors for various times must be calculated to indicate the relationship between the comparative sequences [27–29].

First, the task establishes an original sequence, then an optimal value is selected, and normalized original sequence is as denominator that divides into standard sequences. After that, we acquire an equal weight model and compute the maximal and minimal values among the sequences. Then, we calculate the relational factors to determine the relational order. The equal weight model is set as given in Equation (1) [25] below.

$$\Delta_{0i} = |x_0(k) - x_i(k)|, i = 1, 2, \ldots n \tag{1}$$

$x_i(k)$ means the $i$th compared series and $x_0(k)$ is the reference series. $\Delta_{0i}$ means the absolute difference between the reference series and compared series. The relational factor is set as in Equation (2) below [25]. Typically, $\xi$ is a distinguishing factor which is set to be 0.5 [25]. In general, the value of $x_0(k)$ is assumed to be the same as $x_i(k)$, and the result of the grey relational grade will be set to one, i.e., both series are seen to be highly related. Equation (3) expresses the relational grade, in which $\beta_K$ denotes a weight factor. Finally, the grey relational order in Equation (4) is determined [25].

$$\gamma(x_0(k), x_i(k)) = \frac{\min\limits_{i}\min\limits_{k}\Delta_{0,i}(k) + \xi\max\limits_{i}\max\limits_{k}\Delta_{0,i}(k)}{\Delta_{0i}(k) + \xi\max\limits_{i}\max\limits_{k}\Delta_{0,i}(k)}, i = 1, 2, \ldots n \tag{2}$$

$$\gamma(x_0, x_i) = \sum_{k=1}^{n} \beta_K\gamma(x_0(k), x_i(k)), i = 1, 2, \ldots n \tag{3}$$

$$\gamma(x_0, x_i) \geq \gamma(x_0, x_j) \tag{4}$$

A continuous sequence in GM (1, 1) is expressed as Equation (5) [25,30], in which, $x^{(0)}(k)$ denotes the $k$th element in series $x$. The superscript $^{(0)}$ means the series is the original series. Equation (6) is a mean sequence.

$$X^{(1)}(k) = \sum_{i=1}^{k} x^{(0)}(i), k = 1, 2, \ldots, n \tag{5}$$

$$Z^{(1)}(k) = \frac{X^{(1)}(k) + X^{(1)}(k+1)}{2}, k = 1, 2, \ldots, n-1 \tag{6}$$

In addition, we calculate the $a$ of the development coefficient and the $b$ of the grey active capacity based on the least squares method, where the parameters for $a$ and $b$ are as given in Equations (7) and (8), respectively [25,30].

$$a = \frac{\sum_{k=2}^{n} z^{(1)}(k)\sum_{k=2}^{n} x^{(0)}(k) - (n-1)\sum_{k=2}^{n} z^{(1)}(k)x^{(0)}(k)}{(n-1)\sum_{k=2}^{n}\left[z^{(1)}(k)\right]^2 - \left[\sum_{k=2}^{n} z^{(1)}(k)\right]^2} \tag{7}$$

$$b = \frac{\sum_{k=2}^{n}\left[z^{(1)}(k)\right]^2\sum_{k=2}^{n} x^{(0)}(k) - \sum_{k=2}^{n} z^{(1)}(k)\sum_{k=2}^{n} z^{(1)}(k)x^{(0)}(k)}{(n-1)\sum_{k=2}^{n}\left[z^{(1)}(k)\right]^2 - \left[\sum_{k=2}^{n} z^{(1)}(k)\right]^2} \tag{8}$$

The grey differential equation is in Equation (9) and the solution is in Equation (10) below.

$$x^{(0)}(k) + aZ^{(1)}(k) = b \tag{9}$$

$$\hat{x}^{(1)}(k+1) = \left(x^0(1) - \frac{b}{a}\right) \times e^{-a(k)} + \frac{b}{a} \tag{10}$$

where *k* represents the grey prediction step. Then, we utilize the solution of a GM (1, 1) in Equation (11) to obtain the investigated value, as below [25,30].

$$\hat{x}^{(0)}(k+1) = \hat{x}^{(1)}(k+1) - \hat{x}^{(0)}(k), k - 1, 2, \ldots n. \tag{11}$$

Example for statistical analysis of oil spills of less than 7 tons from 1974 to 2012 is tabulated in the Appendix A.

## 3. Results and Discussion

The experiment utilizes the grey theory to explore oil tanker pollution events caused by various factors from 1974 to 2015 and successfully investigates the causes of events in 2016. Statistical data sourced from the International Tanker Owners Pollution Federation Ltd. is analyzed using grey relational analysis in the process. According to this data, seven types of events cause oil spills from vessels, including collision, grounding, hull failure, equipment failure, fire and explosion, other, and unknown. For historical reasons, spills are generally categorized by size [24], as less than 7 tons (50 barrels), between 7 tons (50 barrels) and 700 tons (5000 barrels), or more than 700 tons (5000 barrels). The details are specified in Table 2.

**Table 2.** Statistics on major oil spills from 1974 to 2015.

| Item | Type | Less Than 7 Tons | Between 7 and 700 Tons | More Than 700 Tons |
|------|------|------------------|------------------------|--------------------|
| 1 | Collision | 188 | 361 | 136 |
| 2 | Grounding | 240 | 270 | 150 |
| 3 | Hull failure | 577 | 101 | 60 |
| 4 | Equipment failure | 1692 | 207 | 18 |
| 5 | Fire/Explosion | 174 | 47 | 52 |
| 6 | * Other | 1815 | 175 | 30 |
| 7 | ** Unknown | 3188 | 203 | 13 |

* Other: Unlisted reasons (e.g., illegal waste oil discharge). ** Unknown: Unknown reasons (e.g., ship sinking).

### 3.1. Analysis of Tanker Oil Spills of Less than 7 Tons

On the basis of the statistical data provided by ITOPF [24], we explored oil tanker pollution caused by four factors (i.e., loading/unloading, bunkering, other, and unknown). Possible events in each factor are specified as follows: Collision, grounding, hull failure, equipment failure, fire/explosion, other reasons, and unknown reasons. Table 3 shows the related data. The colored bars indicate the quantities from 1974 to 2015.

**Table 3.** Statistics on oil spills of less than 7 tons from 2012 to 2015.

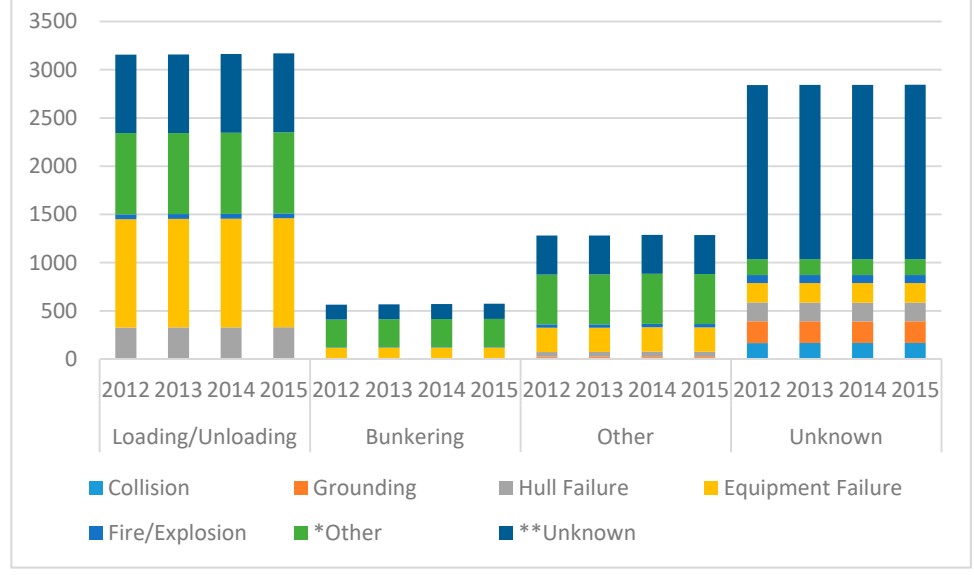

* Other: Unlisted reasons (e.g., illegal waste oil discharge). ** Unknown: Unknown reasons (e.g., ship sinking).

Table 4 shows the result of grey relational analysis and the ranking of oil spills of less than 7 tons using Equation (4). Among the seven types of events (collision, grounding, hull failure, equipment failure, fire or explosion, others, and unknown reasons), oil spills of less than 7 tons occurred mainly for 'other' reasons (i.e., unlisted reasons such as illegal waste oil discharge), as indicated in italic type in the table. A high relational factor indicates a high relational grade.

**Table 4.** Grey relational analysis and ranking of oil spills of less than 7 tons.

| Type | 1974–2012 | Rank | 1974–2013 | Rank | 1974–2014 | Rank | 1974–2015 | Rank |
|---|---|---|---|---|---|---|---|---|
| Collision | 0.341 | 7 | 0.341 | 7 | 0.341 | 7 | 0.501 | 7 |
| Grounding | 0.343 | 5 | 0.342 | 6 | 0.343 | 6 | 0.503 | 6 |
| Hull failure | 0.367 | 4 | 0.367 | 4 | 0.367 | 4 | 0.529 | 4 |
| Equipment Failure | 0.573 | 3 | 0.574 | 3 | 0.574 | 3 | 0.695 | 3 |
| Fire/Explosion | 0.343 | 5 | 0.343 | 5 | 0.344 | 5 | 0.504 | 5 |
| * Other | *0.755* | *1* | *0.755* | *1* | *0.884* | *1* | *0.826* | *1* |
| ** Unknown | 0.714 | 2 | 0.714 | 2 | 0.715 | 2 | 0.820 | 2 |

* Other: Unlisted reasons (e.g., illegal waste oil discharge). ** Unknown: Unknown reasons (e.g., ship sinking).

Table 5 tabulates the error of the calculation according to the grey relational analysis of Equation (11). For oil spills of less than 7 tons, our investigation indicated that 190 collision events, 240 grounding events, 578 hull failure events, 1698 equipment failure events, 175 fire or explosion events, 1817 other events, and 3193 unknown events would occur in 2016, round off after the decimal. By comparing the investigative and actual values, we found that the estimated error was between 0 and 0.09%. The estimated error for collision events was the largest 0.09%. The investigative values are therefore highly accurate and reliable and can serve as a reference for related units to prevent and respond to oil spill events.

**Table 5.** Analysis of results of oil spills of less than 7 tons from 1974 to 2016.

| Item | Collision | Grounding | Hull Failure | Equipment Failure | Fire/Explosion | Other | Unknown |
|---|---|---|---|---|---|---|---|
| T1974–2012 | 182 | 240 | 576 | 1681 | 173 | 1811 | 3178 |
| I1974–2012 | 182 | 240 | 576 | 1681 | 173 | 1811 | 3178 |
| Error (%) | 0.00% | 0.00% | 0.00% | 0.00% | 0.00% | 0.00% | 0.00% |
| T1974–2013 | 185 | 240 | 576 | 1685 | 173 | 1812 | 3179 |
| I1974–2013 | 185 | 240 | 576 | 1685 | 173 | 1812 | 3179 |
| Error (%) | 0.01% | 0.00% | 0.01% | 0.01% | 0.01% | 0.02% | 0. 01% |
| T1974–2014 | 187 | 240 | 577 | 1688 | 174 | 1814 | 3184 |
| I1974–2014 | 187 | 240 | 577 | 1688 | 174 | 1814 | 3184 |
| Error (%) | 0.01% | 0.00% | 0.01% | 0.01% | 0.01% | 0.01% | 0.01% |
| T1974–2015 | 188 | 240 | 577 | 1692 | 174 | 1815 | 3188 |
| I1974–2015 | 188 | 240 | 577 | 1692 | 174 | 1815 | 3188 |
| Error (%) | 0.09% | 0.00% | 0.03% | 0.02% | 0.10% | 0.01% | 0.01% |
| I1974–2016 | 190 | 240 | 578 | 1697 | 175 | 1817 | 3193 |

Remark: 1974–2016, T: True value, I: Investigated value.

### 3.2. Analysis of Tanker Oil Spills of 7–700 Tons

On the basis of the statistical data quoted by ITOPF [24], we next investigated oil spills of between 7 and 700 tons. Table 6 shows the related annual data. As shown in Table 7, the oil spill events of between 7 and 700 tons are mainly caused by collisions and other events. Equipment failure is the third cause. Whether a ship collision or equipment failure occurs, the crew must take emergency action to prevent an oil spill and reduce the disaster.

**Table 6.** Statistics on oil spills of 7–700 tons from 2012 to 2015.

* Other: Unlisted reasons (e.g., illegal waste oil discharge). ** Unknown: Unknown reasons (e.g., ship sinking).

**Table 7.** Grey relational analysis and ranking of oil spills of 7-700 tons.

| Type | 1974–2012 | Rank | 1974–2013 | Rank | 1974–2014 | Rank | 1974–2015 | Rank |
|---|---|---|---|---|---|---|---|---|
| Collision | 0.668 | 2 | *0.669* | *1* | *0.669* | *1* | *0.669* | *1* |
| Grounding | 0.485 | 5 | 0.476 | 5 | 0.475 | 5 | 0.475 | 5 |
| Hull failure | 0.399 | 6 | 0.4 | 6 | 0.401 | 6 | 0.401 | 6 |
| Equipment failure | 0.576 | 3 | 0.569 | 3 | 0.567 | 3 | 0.567 | 3 |
| Fire/Explosion | 0.366 | 7 | 0.359 | 7 | 0.358 | 7 | 0.358 | 7 |
| * Other | *0.687* | *1* | 0.646 | 2 | 0.643 | 2 | 0.643 | 2 |
| ** Unknown | 0.523 | 4 | 0.518 | 4 | 0.517 | 4 | 0.517 | 4 |

* Other: Unlisted reasons (e.g., illegal waste oil discharge). ** Unknown: Unknown reasons (e.g., ship sinking).

According to grey modelling, the expected frequencies of future events can be investigated. Table 7 shows the result of grey relational analysis and the ranking of oil spills of between 7 and 700 tons using Equation (4). Table 8 indicates the results of oil spills of 7-700 tons from 1974 to 2016, where the numbers of collision events, grounding events, hull failure events, equipment failure events, fire or explosion events, other events, and unknown events in 2016 are 364, 270, 102, 209, 48, 177, and 205, respectively, round off after the decimal. By comparing the investigative and actual values, we found that the estimated error is between 0 and 0.64%; the estimated error for hull failure events is the largest 0.64%. This is because of equipment failure related to the main engine, auxiliary machines, and generators. For example, De Xiang Taipei lost power and was grounded because of equipment failure; it then broke into two parts, damaging its fuel tank, which resulted in an oil spill and caused severe pollution [31]. According to the results, the estimated grades are highly accurate and reliable, and the estimated error is only 0.64%.

**Table 8.** Analysis of results of oil spills of 7-700 tons from 1974 to 2016.

| Item | Collision | Grounding | Hull Failure | Equipment Failure | Fire/Explosion | Other | Unknown |
|---|---|---|---|---|---|---|---|
| $^T$1974–2012 | 350 | 270 | 99 | 203 | 45 | 169 | 206 |
| $^I$1974–2012 | 350 | 270 | 99 | 203 | 45 | 169 | 206 |
| Error (%) | 0.00% | 0.00% | 0.00% | 0.00% | 0.00% | 0.00% | 0.00% |
| $^T$1974–2013 | 354 | 270 | 100 | 203 | 46 | 171 | 206 |
| $^I$1974–2013 | 354 | 270 | 100 | 203 | 46 | 171 | 206 |
| Error (%) | 0. 02% | 0.00% | 0.08% | 0.04% | 0.18% | 0.05% | 0.00% |
| $^T$1974–2014 | 355 | 270 | 101 | 204 | 47 | 172 | 206 |
| $^I$1974–2014 | 355 | 270 | 101 | 204 | 47 | 172 | 206 |
| Error (%) | 0.01% | 0.00% | 0.01% | 0.01% | 0.01% | 0.01% | 0.00% |
| $^T$1974–2015 | 361 | 270 | 101 | 207 | 47 | 175 | 206 |
| $^I$1974–2015 | 360 | 270 | 100 | 207 | 47 | 175 | 206 |
| Error (%) | 0.22% | 0.00% | 0.64% | 0.16% | 0.35% | 0.18% | 0.24% |
| $^I$1974–2016 | 364 | 270 | 102 | 209 | 48 | 177 | 205 |

Remark: 1974-2016, T: True value, I: Investigated value.

## 3.3. Analysis of Tanker Oil Spills Greater than 700 tons

On the basis of the statistical data quoted by ITOPF [24] oil spills of more than 700 tons are detailed in Table 9.

**Table 9.** Statistics for oil spills over 700 tons from 2012 to 2015.

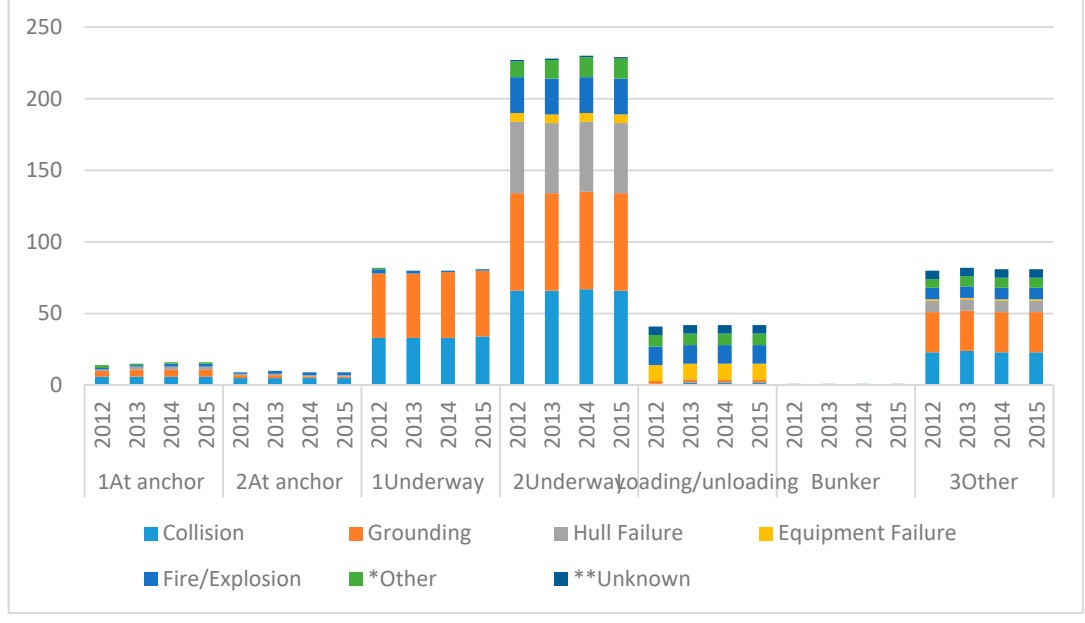

* Other: Unlisted reasons (e.g., illegal waste oil discharge). ** Unknown: Unknown reasons (e.g., ship sinking).

Among the seven types of oil spill events in Table 10, oil spills greater than 700 tons were mainly caused by collisions. Table 11 tabulates the numbers of collision events, grounding events, hull failure events, equipment failure events, fire or explosion events, other events, and unknown events in 2016, which are analyzed to be 137, 151, 60, 18, 52, 31, and 13, respectively, round off after the decimal. By comparing the estimated and actual values, the estimated error is between 0 and 3.43%; the estimated error for 'other' events is the largest 3.43%. The estimated grade is highly accurate and reliable.

**Table 10.** Grey relational analysis and ranking of oil spills over 700 tons.

| Type | 1974–2012 | Rank | 1974–2013 | Rank | 1974–2014 | Rank | 1974–2015 | Rank |
|---|---|---|---|---|---|---|---|---|
| Collision | *0.717* | *1* | *0.726* | *1* | *0.722* | *1* | *0.720* | *1* |
| Grounding | 0.680 | 2 | 0.701 | 2 | 0.691 | 2 | 0.691 | 2 |
| Hull failure | 0.404 | 4 | 0.409 | 4 | 0.409 | 4 | 0.409 | 4 |
| Equipment failure | 0.399 | 5 | 0.399 | 5 | 0.399 | 5 | 0.399 | 5 |
| Fire/Explosion | 0.568 | 3 | 0.575 | 3 | 0.582 | 3 | 0.582 | 3 |
| * Other | 0.394 | 6 | 0.389 | 6 | 0.390 | 6 | 0.390 | 6 |
| ** Unknown | 0.363 | 7 | 0.364 | 7 | 0.363 | 7 | 0.363 | 7 |

* Other: Unlisted reasons (e.g., illegal waste oil discharge). ** Unknown: Unknown reasons (e.g., ship sinking).

The next task is to carry out grey relational analysis to identify the development trend based on the closeness of the curve shapes of pairs of factor sequences. If two factors present a consistent development trend during their systematic development, then they are highly correlated with each other; otherwise, the two factors are insignificantly correlated with each other. Therefore, grey relational analysis is suitable for forecasting the causes of oil tanker spills.

Grey relational analysis was applied on oil spills of less than 7 tons, between 7 and 700 tons, and more than 700 tons. The result shows that the investigative accuracy reached 96%.

**Table 11.** Analysis of results of oil spills over 700 tons from 1974 to 2016.

| Item | Collision | Grounding | Hull Failure | Equipment Failure | Fire/Explosion | Other | Unknown |
|---|---|---|---|---|---|---|---|
| $^T$1974–2012 | 134 | 149 | 60 | 18 | 52 | 28 | 13 |
| $^I$1974–2012 | 134 | 149 | 60 | 18 | 52 | 28 | 13 |
| Error (%) | 0.00% | 0.00% | 0.00% | 0.00% | 0.00% | 0.00% | 0.00% |
| $^T$1974–2013 | 136 | 150 | 60 | 18 | 52 | 29 | 13 |
| $^I$1974–2013 | 136 | 150 | 60 | 18 | 52 | 29 | 13 |
| Error (%) | 0.01% | 0.01% | 0.00% | 0.00% | 0.00% | 0.01% | 0.00% |
| $^T$1974–2014 | 136 | 150 | 60 | 18 | 52 | 30 | 13 |
| $^I$1974–2014 | 136 | 150 | 60 | 18 | 52 | 30 | 13 |
| Error (%) | 0.01% | 0.01% | 0.00% | 0.00% | 0.00% | 0.01% | 0.00% |
| $^T$1974–2015 | 136 | 150 | 60 | 18 | 52 | 30 | 13 |
| $^I$1974–2015 | 137 | 151 | 60 | 18 | 52 | 31 | 13 |
| Error (%) | 0.74% | 0.67% | 0.00% | 0.00% | 0.00% | 3.43% | 0.00% |
| $^I$1974–2016 | 137 | 151 | 60 | 18 | 52 | 31 | 13 |

Remark: 1974–2016, T: true value, I: investigated value.

### 3.4. Discussion and Summary

Grey theory was applied for the investigation of three types of oil spill events, as shown in Table 5, Table 8, and Table 11. The details are as follows:

(1) For oil spills of less than 7 tons, the largest difference between the investigative and true values was for collision events and the error of the investigative value was 0.09% compared with the true value, as shown in Table 5 from 1974 to 2016.

(2) For oil spills of between 7 and 700 tons, the largest difference between the investigative and true values was for hull failure events and the error of the investigative value was 0.64% compared with the true value, as shown in Table 8 from 1974 to 2016.

(3) For oil spills of more than 700 tons, the largest difference between the investigative and true values was 'other' events and the error of the investigative value was 3.43% compared with the true value, as shown in Table 11 from 1974 to 2016.

In the present study, the experiment was able to identify the factors causing oil spill events and to assess the relationships between various factors. We ranked the factors to determine their relationships; in addition, based on the influences of the grey relations, we found that the differences between the true values and investigative values were all less than 3.43%. Consequently, the investigation is highly accurate. The paper utilized one deals with the low volume incidents to approach the accident analysis of oil spill. The verification is quite satisfying to verify the cause of oil spill for oil tanker that started from 1974 to 2015 in the past. Therefore, a simple application of grey theory can be fast tool and reliable strategy to decision maker whom prevent the oil spill from the ocean in future.

### 4. Conclusions

An empirical study was performed using grey theory for statistical analysis of oil spill data from ITOPF from 1974 to 2015 and for an investigation of the distribution of events in 2016. The results are very consistent, with an accuracy that is greater than 96% and the error is within 3.43%. Oil spill events of less than 7 tons are mainly caused by 'other' factors, while those of between 7 and 700 tons and of more than 700 tons are mainly caused by collisions. The result of the research is a significant benchmark of relevance to multiple ship types, cargo types, and ship structures concerning the various types of oil spill events and can contribute to formulating policies for preventing marine pollution. In the marine environment, oil exploration and mining activities as well as commercial shipping activities are at risk of causing oil pollution. Thus, an adequate management is required to prevent and respond to oil spill events.

**Author Contributions:** Data curation, C.H.C.; Formal analysis, C.H.C.; Methodology, C.H.C., F.M.T.; Validation, D.T.S., F.M.T.; writing—original draft, F.M.T.; Writing—review & editing, F.M.T. and D.T.S.

**Funding:** This research received no external funding.

**Conflicts of Interest:** The authors declare no conflicts of interest.

## Appendix A

**Table A1.** Results of oil spill less than 7 tons from 1974 to 2012.

| Type | Operation (Loading/Discharging) | Operation (Bunkering) | Operation (Other) | Operation (Unknown) | Total |
|---|---|---|---|---|---|
| Collision | 1 | 2 | 13 | 166 | 182 |
| Grounding | 2 | 0 | 14 | 226 | 242 |
| Hull Failure | 324 | 10 | 47 | 195 | 577 |
| Equipment Failure | 1124 | 104 | 251 | 202 | 1681 |
| Fire/Explosion | 50 | 5 | 35 | 83 | 173 |
| * Other | 842 | 289 | 517 | 163 | 1811 |
| ** Unknown | 814 | 154 | 404 | 1806 | 3178 |

* Other: Unlisted reasons (e.g., illegal waste oil discharge). ** Unknown: Unknown reasons (e.g., ship sinking).

At first, we take the standardize from raw data of Table A1, and use the reference series as the benchmark for data analysis by formula, $x_i(k) = \frac{\min[x_i(k)]}{\max[x_i(k)]}$, acquires the standardization of the series, throughout $X_1$–$X_7$, accordingly.

**Table A2.** Standardization of the series.

| Series and Factor (K) | K = 1 | K = 2 | K = 3 | K = 4 |
|---|---|---|---|---|
| $X_0$ | 1 | 1 | 1 | 1 |
| $X_1$ | 0.001 | 0.007 | 0.025 | 0.092 |
| $X_2$ | 0.002 | 0 | 0.027 | 0.125 |
| $X_3$ | 0.288 | 0.035 | 0.091 | 0.109 |
| $X_4$ | 1 | 0.36 | 0.485 | 0.112 |
| $X_5$ | 0.044 | 0.017 | 0.068 | 0.046 |
| $X_6$ | 0.749 | 1 | 1 | 0.09 |
| $X_7$ | 0.724 | 0.533 | 0.781 | 1 |

1. The raw data has met the comparability so the original data is used for gray correlation analysis.
2. $\Delta_{0i}(k) = |x_0(k) - x_i(k)|$ as Equation (1) we can calculate the result as below, which i = 1, 2, ... , 7, k = 1, 2, ... ,

A.  $\Delta_{01}(1) = 0.9991 \; \Delta_{01}(2) = 0.9931 \; \Delta_{01}(3) = 0.9749 \; \Delta_{01}(4) = 0.9081$
B.  $\Delta_{02}(1) = 0.9982 \; \Delta_{02}(2) = 1.0000 \; \Delta_{02}(3) = 0.9729 \; \Delta_{02}(4) = 0.8749$
C.  $\Delta_{03}(1) = 0.7117 \; \Delta_{03}(2) = 0.9654 \; \Delta_{03}(3) = 0.9091 \; \Delta_{03}(4) = 0.8915$
D.  $\Delta_{04}(1) = 0.0000 \; \Delta_{04}(2) = 0.6401 \; \Delta_{04}(3) = 0.5145 \; \Delta_{04}(4) = 0.8882$
E.  $\Delta_{05}(1) = 0.9555 \; \Delta_{05}(2) = 0.9827 \; \Delta_{05}(3) = 0.9323 \; \Delta_{05}(4) = 0.9540$
F.  $\Delta_{06}(1) = 0.2509 \; \Delta_{06}(2) = 0.0000 \; \Delta_{06}(3) = 0.0000 \; \Delta_{06}(4) = 0.9097$
G.  $\Delta_{07}(1) = 0.2758 \; \Delta_{07}(2) = 0.4671 \; \Delta_{07}(3) = 0.2186 \; \Delta_{07}(4) = 0.0000$

Result as below

$\Delta_{01} = (\; 0.9991, \; 0.9931, \; 0.9749, \; 0.9081 \;)$
$\Delta_{02} = (\; 0.9982, \; 1.0000, \; 0.9729, \; 0.8749 \;)$
$\Delta_{03} = (\; 0.7117, \; 0.9654, \; 0.9091, \; 0.8915 \;)$

$\Delta_{04} = ( \ 0.0000, \ 0.6401, \ 0.5145, \ 0.8882 \ )$

$\Delta_{05} = ( \ 0.9555, \ 0.9827, \ 0.9323, \ 0.9540 \ )$

$\Delta_{06} = ( \ 0.2509, \ 0.0000, \ 0.0000, \ 0.9097 \ )$

$\Delta_{07} = ( \ 0.2758, \ 0.4671, \ 0.2186, \ 0.0000 \ )$

3.  Find the maximum difference and the minimum difference between the two poles

$\Delta_{max} = |x_0(k) - x_1(k)| = \Delta_{01} \ (4) = 1.00$

$\Delta_{max} = |x_0(k) - x_2(k)| = \Delta_{02} \ (4) = 1.00$

$\Delta_{max} = |x_0(k) - x_3(k)| = \Delta_{03} \ (4) = 0.97$

$\Delta_{max} = |x_0(k) - x_4(k)| = \Delta_{04} \ (4) = 0.89$

$\Delta_{max} = |x_0(k) - x_5(k)| = \Delta_{05} \ (4) = 0.98$

$\Delta_{max} = |x_0(k) - x_6(k)| = \Delta_{06} \ (4) = 0.91$

$\Delta_{max} = |x_0(k) - x_7(k)| = \Delta_{07} \ (4) = 0.47$

$\Delta_{min} = |x_0(k) - x_1(k)| = \Delta_{01} \ (1) = 0.91$

$\Delta_{min} = |x_0(k) - x_2(k)| = \Delta_{02} \ (1) = 0.87$

$\Delta_{min} = |x_0(k) - x_3(k)| = \Delta_{03} \ (1) = 0.71$

$\Delta_{min} = |x_0(k) - x_4(k)| = \Delta_{04} \ (1) = 0.00$

$\Delta_{min} = |x_0(k) - x_5(k)| = \Delta_{05} \ (1) = 0.93$

$\Delta_{min} = |x_0(k) - x_6(k)| = \Delta_{06} \ (1) = 0.00$

$\Delta_{min} = |x_0(k) - x_7(k)| = \Delta_{07} \ (1) = 0.00$

The difference at max.is 1.00, the difference at min.is 0.00

4.  Set $\zeta$(Zeta) distinguishing factor = 0.5

5.  Calculate the grey correlation coefficient: use Equation (2) to find the value as below,

$\gamma \ ( \ x_0(k), \ x_i(k)) = \frac{0 + \ (0.5)*(1.00)}{\Delta_{0i}(k) + \ (0.5)*(1.00)}$, consequently as below.

H.  $\gamma(x_0(1), x_1(1)) = 0.334 \ \gamma(x_0(2), x_1(2)) = 0.335$
$\gamma(x_0(3), x_1(3)) = 0.339 \ \gamma(x_0(4), x_1(4)) = 0.355$

I.  $\gamma(x_0(1), x_2(1)) = 0.334 \ \gamma(x_0(2), x_2(2)) = 0.333$
$\gamma(x_0(3), x_2(3)) = 0.339 \ \gamma(x_0(4), x_2(4)) = 0.364$

J.  $\gamma(x_0(1), x_3(1)) = 0.413 \ \gamma(x_0(2), x_3(2)) = 0.341$
$\gamma(x_0(3), x_3(3)) = 0.355 \ \gamma(x_0(4), x_3(4)) = 0.359$

K.  $\gamma(x_0(1), x_4(1)) = 1.000 \ \gamma(x_0(2), x_4(2)) = 0.439$
$\gamma(x_0(3), x_4(3)) = 0.493 \ \gamma(x_0(4), x_4(4)) = 0.360$

L.  $\gamma(x_0(1), x_5(1)) = 0.344 \ \gamma(x_0(2), x_5(2)) = 0.337$
$\gamma(x_0(3), x_5(3)) = 0.349 \ \gamma(x_0(4), x_5(4)) = 0.344$

M.  $\gamma(x_0(1), x_6(1)) = 0.666 \ \gamma(x_0(2), x_6(2)) = 1.000$
$\gamma(x_0(3), x_6(3)) = 1.000 \ \gamma(x_0(4), x_6(4)) = 0.355$

N.  $\gamma(x_0(1), x_7(1)) = 0.644 \ \gamma(x_0(2), x_7(2)) = 0.517$
$\gamma(x_0(3), x_7(3)) = 0.696 \ \gamma(x_0(4), x_7(4)) = 1.000$

6.  Calculate the gray correlation degree: subject to the equal weight as Equations (3) and (4)

$\gamma(x_0, x_i) = \sum_{k=1}^{m} \beta_k \gamma(x_0(k), \ x_i(k))$, result as below,

Value of $\gamma(x_0, x_1) \ \gamma(x_0, x_1) = \frac{1}{4} (0.334 + 0.335 + 0.339 + 0.355) = 0.341$

Value of $\gamma(x_0, x_2) \ \gamma(x_0, x_2) = \frac{1}{4} (0.334 + 0.333 + 0.339 + 0.364) = 0.343$

Value of $\gamma(x_0, x_3) \ \gamma(x_0, x_3) = \frac{1}{4} (0.413 + 0.341 + 0.355 + 0.359) = 0.367$

Value of $\gamma(x_0, x_4) \ \gamma(x_0, x_4) = \frac{1}{4} (1.000 + 0.439 + 0.493 + 0.360) = 0.573$

Value of $\gamma(x_0, x_5) \ \gamma(x_0, x_5) = \frac{1}{4} (0.344 + 0.337 + 0.349 + 0.344) = 0.343$

Value of $\gamma(x_0, x_6) \gamma(x_0, x_6) = \frac{1}{4} (0.666 + 1.000 + 1.000 + 0.355) = 0.755$

Value of $\gamma(x_0, x_7) \ \gamma(x_0, x_7) = \frac{1}{4} (0.644 + 0.517 + 0.696 + 1.000) = 0.714$

**Table A3.** Statistical analysis of oil spills of less than 7 tons from 1974 to 2012.

| Standardization | | | | |
|---|---|---|---|---|
| **Type** | Operation (Loading/Unloading) | Operation (Bunkering) | Operation (Other) | Operation (Unknown) |
| Collision | 0.001 | 0.007 | 0.025 | 0.092 |
| Grounding | 0.002 | 0.000 | 0.027 | 0.125 |
| Hull Failure | 0.288 | 0.035 | 0.091 | 0.109 |
| Equipment Failure | 1.000 | 0.360 | 0.485 | 0.112 |
| Fire/Explosion | 0.044 | 0.017 | 0.068 | 0.046 |
| * Other | 0.749 | 1.000 | 1.000 | 0.09 |
| ** Unknown | 0.724 | 0.533 | 0.781 | 1.000 |
| Difference sequences | | | | |
| **Type** | Operation (Loading/Unloading) | Operation (Bunkering) | Operation (Other) | Operation (Unknown) |
| Collision | 0.999 | 0.993 | 0.975 | 0.908 |
| Grounding | 0.998 | 1.000 | 0.973 | 0.875 |
| Hull Failure | 0.712 | 0.965 | 0.909 | 0.892 |
| Equipment Failure | 0.000 | 0.640 | 0.515 | 0.888 |
| Fire/Explosion | 0.956 | 0.983 | 0.932 | 0.954 |
| * Other | 0.251 | 0.000 | 0.000 | 0.909 |
| ** Unknown | 0.276 | 0.467 | 0.217 | 0.000 |
| Factor | | | | |
| **Type** | Operation (Loading/Unloading) | Operation (Bunkering) | Operation (Other) | Operation (Unknown) |
| Collision | 0.334 | 0.335 | 0.339 | 0.355 |
| Grounding | 0.334 | 0.333 | 0.339 | 0.364 |
| Hull Failure | 0.413 | 0.341 | 0.355 | 0.359 |
| Equipment Failure | 1.000 | 0.439 | 0.493 | 0.360 |
| Fire/Explosion | 0.344 | 0.337 | 0.349 | 0.344 |
| * Other | 0.666 | 1.000 | 1.000 | 0.355 |
| ** Unknown | 0.644 | 0.517 | 0.696 | 1.000 |

| Relationship | | | | | | | |
|---|---|---|---|---|---|---|---|
| **Item** | Collision | Grounding | Hull Failure | Equipment Failure | Fire/Explosion | Other | Unknown |
| Factor | 0.341 | 0.343 | 0.367 | 0.573 | 0.343 | *0.755* | 0.714 |
| Rank | 7 | 5 | 4 | 3 | 5 | *1* | 2 |

* Other: Unlisted reasons (e.g., illegal waste oil discharge). ** Unknown: Unknown reasons (e.g., ship sinking).

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
