# Peer review of "Investigation of Oil Spills from Oil Tankers through Grey Theory: Events from 1974 to 2016"

_jmse, doi:10.3390/jmse7100373_

Round 1

Reviewer 1 Report

The authors need to correct the significant figures throughout the whole discussion to 3 or less

The authors need to discuss how one deals with the low volume incidents where the majority of causality is unknown - doesn't this affect the reliability of results

Author Response

Point-by-point reply to Reviewers’ comments and suggestions.

Dear Editor,

The authors greatly appreciate reviewers’ comments and suggestions to our paper. The paper has been revised accordingly. Major revisions are colorful; those for Reviewer 1 are marked in red, Reviewer 2 are green and those for Reviewer 3 in blue for ready reference. Details of the point-by-point replies are as follows.

Sincerely,

Fu-Ming Tzu

To Reviewer 1 response.

Point 1: The authors need to correct the significant figures throughout the whole discussion to 3 or less

Reply 1: Thanks for Reviewer the valuable comment and advice.

Revision, accordingly. The authors amend to take the third decimal point throughout the discussions.

Point 2: The authors need to discuss how one deals with the low volume incidents where the majority of causality is unknown - doesn't this affect the reliability of results

Reply 2: The authors expresses the appreciation to reviewer’s advice. The response is as below and the authors also supply the comment in 3.4 Discussion and Summary.

The paper utilized one deals with the low volume incidents to approach the accident analysis of oil spill. The verification is quite satisfying to verify the cause of oil spill for oil tanker that started from 1974 to 2015 in the past. Therefore, a simple application of grey theory can be fast tool and reliable strategy to decision maker whom prevent the oil spill from the ocean in future.

Besides, the meaning of unknown is a suspended circumstance e.g., the ship sinking and the ship is under legal investigation (less than 7 tonnes for oil spill).

Point 3: I would suggest that since unknown and other are the largest amounts that this data may not be useful for this purpose

Reply 3: Many thanks the reviewer reminding. The authors are very apology to lead unclear message so that the authors shall correct to add the information in the paper, as below.

*Other: unlisted reasons (e.g., illegal waste oil discharge).

**Unknown: unknown reasons (e.g., ship sinking).

Since both other and unknown are the majority of oil spill which is less than 7 tonnes, small quantity of oil spill is sometime caused by many reasons. However, one of the reason might be illegal oil discharge. Moreover, unknown is a suspended but it might be caused by ship sinking. Both causes are useful and verified by ITOPF. Thus, the authors suggest to remain the style.

Point 4: way too many significant figures

Reply 4: The authors amend to take the third decimal point throughout the discussions.

Reviewer 2 Report

The presentation of the topic and the research is very good from both points of view: Theoretical and Statistical results. The way of progress is between both previous points. The lack of relation between the theory and the results needs more details, like for example the values of γ(xi,xj) in the results or the usage of the theoretical notations (a, b, k…) in the different tables.

The relevance of the Grey theory is clear but there are insufficient details about the irrelevance of others classical statistical methods.

The link between the differential equation and the ranges of several period extensions from 2012 for the three cases if exists must be detailed. It may explain more why the columns rank of table 10 for example remain the same.

The statistical analysis concerns ITOPF data solely and we can find others sources of data/information by example (A. Zafirakou et. al., 2018 J. Mar. Sci. Eng. 6(4) 125) about ship collision causing oil-spill, including Exxon Valdez and Atlantic Express.

The title must indicate that the investigation is about past events.

Check 0.09% or 0.10 % L 203

Author Response

To Reviewer 2 response.

Point 1: The presentation of the topic and the research is very good from both points of view: Theoretical and Statistical results. The way of progress is between both previous points. The lack of relation between the theory and the results needs more details, like for example the values of γ(xi,xj) in the results or the usage of the theoretical notations (a, b, k…) in the different tables.

Reply 1: The authors expresses the appreciation to reviewer’s advice. Herewith the enclosed is the detailed calculation to present the values of γ (xi, xj) and standardization, difference sequence, relationship factor and relationship, respectively, in appendix 2. Example for the analysis of results of oil spill less than 7 tons from 1974 to 2012 is in the appendix 1, as below.

Table A1: Results of oil spill less than 7 tons from 1974 to 2012

Operation
(Loading/Discharging)

Operation (Bunkering)

Operation
 (Other)

Operation (Unknown)

Total

Collision

1

2

13

166

182

Grounding

2

0

14

226

242

Hull Failure

324

10

47

195

577

 Equipment Failure

1124

104

251

202

1681

 Fire/Explosion

50

5

35

83

173

 Other

842

289

517

163

1811

 Unknown

814

154

404

1806

3178

At first, we take the standardization from raw data of Table A1, and we use the reference series as the benchmark for data analysis by formula, , acquires the standardization of the series, throughout X1~X7 , accordingly.   

Table A2: Standardization of the series

Series & Factor (K)

K = 1

K = 2

K = 3

K = 4

X0

1

1

1

1

X1

0.001

0.007

0.025

0.092

X2

0.002

0

0.027

0.125

X3

0.288

0.035

0.091

0.109

X4

1

0.36

0.485

0.112

X5

0.044

0.017

0.068

0.046

X6

0.749

1

1

0.09

X7

0.724

0.533

0.781

1

The raw data has met the comparability so the original data is used for gray correlation analysis. (k) = as Eq. (1),we can calculate the result as below。which i = 1, 2, …, 7,k = 1, 2, …, , 

   ,

, 

   ,

,

   ,

,

   ,

,

   ,

,

   ,

,

   ,

Result as below

Find the maximum difference and the minimum difference between the two poles

 =

 =

 =

 =

 =

 =

 =

 =

 =

 =

 =

 =

 =

 =

The difference at max.is 1.00, the difference at min.is 0.00。

Set ζ(Zeta) distinguishing factor = 0.5 Calculate the grey correlation coefficient: use Eq. (2) to find the value as below,

   , consequently as below,

= 0.334   , = 0.335

 = 0.339   , = 0.355

= 0.334   , = 0.333

 = 0.339   , = 0.364

= 0.413   , = 0.341

 = 0.355   , = 0.359

= 1.000   , = 0.439

 = 0.493   ,= 0.360

= 0.344   , = 0.337

 = 0.349   , = 0.344

= 0.666   , = 1.000

 = 1.000   , = 0.355

= 0.644   , = 0.517

 = 0.696   , = 1.000

Calculate the gray correlation degree: subject to the equal weight as Eq. (3) and Eq. (4)

) = , result as below,

) ,) =  ( 0.334 + 0.335 + 0.339 + 0.355 ) = 0.341

) ,) =  ( 0.334 + 0.333 + 0.339 + 0.364 ) = 0.343

) ,) =  ( 0.413 + 0.341 + 0.355 + 0.359 ) = 0.367

) ,) =  ( 1.000 + 0.439 + 0.493 + 0.360 ) = 0.573

) ,) =  ( 0.344 + 0.337 + 0.349 + 0.344 ) = 0.343

),) =  ( 0.666 + 1.000 + 1.000 + 0.355) = 0.755

) ,) =  ( 0.644 + 0.517 + 0.696 + 1.000 ) = 0.714

Point 2: The relevance of the Grey theory is clear but there are insufficient details about the irrelevance of others classical statistical methods. The link between the differential equation and the ranges of several period extensions from 2012 for the three cases if exists must be detailed. It may explain more why the columns rank of table 10 for example remain the same.

Reply 2: Many thanks for reviewer advice. Since the research are not focused on the irrelevance of others classical statistical methods so that the authors does not specify in detailed.

We heard that the irrelevance of classical statistics can support the analysis more realistic and reliable. Nevertheless, the grey relational analysis (GRA) has performed in multivariate time series. This is, GRA is a simple and accurate method in various disciplines such as engineering, economics, politics and receives good reputation. Whereas, the authors are pretty appreciations toward the valuable suggests from reviewer. The authors will take the consideration at next research.

Besides, we add the supplement in appendix 2 to describe the detailed calculation with respect to three case of oil spill between 2012 and 2015 started in1974.

Point 3: The statistical analysis concerns ITOPF data solely and we can find others sources of data/information by example (A. Zafirakou et. al., 2018 J. Mar. Sci. Eng. 6(4) 125) about ship collision causing oil-spill, including Exxon Valdez and Atlantic Express.

Reply 3: The authors expresses the appreciation to reviewer’s advice. The information from reviewer have added to the literature review and reference.

Zafirakou et al. utilized “multi-criteria analysis of different approaches to protect the marine and coastal environment from oil spills” to discuss the marine pollution caused by oil spill of oil tanker. Taking the responsibility from local authorities and the relevant unit in case of an oil spill accident, through a multi-criteria methodology.

Point 4: The title must indicate that the investigation is about past events.

Reply 4: Many thanks for reviewer advice. The title revise as below. 

Investigation of Oil Spills from Oil Tankers through Grey Theory: Event from 1974 to 2016

Point 5: Check 0.09% or 0.10 % L 203

Reply 5: Thanks for the advice. It is 0.09%.

Reviewer 3 Report

The manus provides an interesting overview of marine oil spills related to ship accidents during the last 40 years.

The authors need to explain more clearly why this study is important. How can we utilize the results specifically to reduce the number of accidents in the future?

Specific comments:

L43: "Meanwhile, extreme weather has frequently increased the
intensity of tropical storms such as typhoons, hurricanes and cyclones."

Do you have references for this statement? Is there an increasing trend of extreme weather in all regions?

Author Response

To Reviewer 3 response.

Point 1: The manus provides an interesting overview of marine oil spills related to ship accidents during the last 40 years. The authors need to explain more clearly why this study is important. How can we utilize the results specifically to reduce the number of accidents in the future?

Reply 1: The authors expresses the appreciation to reviewer’s advice. Additional part is in the content.

The research is aimed to reduce the marine accidents and predict the cause of oil spill in advance. The analysis is dealing with the incidents to approach the circumstance in various intensity of oil spill in the last 40 years. Moreover, an application of grey theory demonstrates accurate and reliable methodology to decision maker. Thus, the investigation can predict the causes of pollution from oil spill accidents in the future.

Specific comments:

Point 2: L43: "Meanwhile, extreme weather has frequently increased the intensity of tropical storms such as typhoons, hurricanes and cyclones." Do you have references for this statement? Is there an increasing trend of extreme weather in all regions?

Reply 2: Many thanks for reviewer advice. Sure, there are several papers to discuss about extreme weather, as result the climate change is more severity and number of natural events have been increasing trend [1-3]. Moreover, Peter et al., discussed the heat and rainfall extremes have intensified over the past few decades and this trend is projected to continue with future global warming [4]. In the end, the authors are very appreciated the valuable suggestions from the reviewer and the paper is also added to the references, as below.

Khomami M. S., Kenari M. T., Sepasian M. S. A warning indicator for distribution network to extreme weather events. International Transactions on Electrical Energy Systems 2019, 29, Fakour H., Lo S. L., Lin T. F. Impacts of Typhoon Soudelor (2015) on the water quality of Taipei, Taiwan. Scientific Reports 2016, 6, Mouche A., Chapron B., Knaff J., Zhao Y. L., Zhang B., Combot C. Copolarized and Cross-Polarized SAR Measurements for High-Resolution Description of Major Hurricane Wind Structures: Application to Irma Category5 Hurricane. Journal of Geophysical Research-Oceans 2019, 124, 3905-3922 Pfleiderer P., Schleussner C. F., Kornhuber K., Coumou D. Summer weather becomes more persistent in a 2 degrees C world. Nature Climate Change 2019, 9, 666-+

Round 2

Reviewer 1 Report

needs editorial changes - a large number of errors were added in the last revision

significant figures need revision throughout

grammatical errors

Author Response

Point-by-point reply to Reviewers’ comments and suggestions

Dear Editor,

The authors greatly appreciate reviewer comments and suggestions to our paper. The paper has been revised accordingly. Minor revisions are red color for ready reference. Details of the point-by-point replies are as follows.

Sincerely,

Fu-Ming Tzu

To Reviewer 1 response.

Point 1: needs editorial changes - a large number of errors were added in the last revision

Reply 1: The authors expresses the appreciation to reviewer advice. An editorial change in this version has corrected.

Point 2: significant figures need revision throughout

Reply 2: Many thanks for the reviewer advice. The authors take a round off after the decimal.

Point 3: grammatical errors

Reply 3: Revision, accordingly. Events from 1974 to 2016
